# Prevalence of uremic neuropathy and the effect of dialysis in children with end-stage renal disease: A cohort study

Arwa Yahyaoui [1][*], Nouha Gammoudi[1,2], Selsabil Nouir[3,4], Sameh Mabrouk[3,4], Hela Ghali[5], Saoussen Abroug[3,4], Ghazi Sakly[1,2]

**1** Neurophysiology Department, Sahloul University Hospital, Sousse, Tunisia, **2** Medical Technology and Image Processing Laboratory, Faculty of Medicine, University of Monastir, Monastir, Tunisia, **3** Pediatric Department, Sahloul University Hospital, Sousse, Tunisia, **4** Faculty of Medicine of Sousse, University of Sousse, Sousse, Tunisia, **5** Department of Preventive and Community Medicine, Sahloul University Hospital, Faculty of Medicine of Sousse, University of Sousse, Sousse, Tunisia

☉ These authors contributed equally to this work.
* arwayahyaoui.ay@gmail.com

## Abstract

Children with chronic kidney disease (CKD) face increased morbidity, mortality, and reduced quality of life. Uremic neuropathy (UN) is a common neurological complication, but data on its relationship with dialysis in pediatric populations are limited. This prospective study aimed to assess the prevalence of UN in children with end-stage renal disease (ESRD) in a Tunisian population and explore the association between dialysis and UN. Conducted between July and September 2023 in the nephrology and neurophysiology units of a Tunisian hospital, the study included 31 children with CKD G5. Clinical data, biological analyses, and nerve conduction studies via electroneuromyography (EMG) were performed at baseline and six months later. Participants were divided into pre-dialysis and dialysis groups for comparison. The mean age was 11 ± 3.5 years, and the average age at CKD diagnosis was 7.5 ± 4.2 years. UN was diagnosed in 45% of participants using EMG, including 13% with silent neuropathy. Axonal neuropathy was predominant, with no cases of demyelinating neuropathy identified. Initial comparisons between dialysis and pre-dialysis groups showed no significant differences in UN characteristics. However, clinical neuropathy, weight-for-age, and glomerular nephritis were significantly associated with UN. Follow-up revealed a significant improvement in UN in the dialysis group. From this study, we conclude the importance of screening for UN in pediatric ESRD care and recommend routine EMG evaluations, even in asymptomatic patients, to ensure early diagnosis and management.

**Data availability statement:** All relevant data are within the paper and its Supporting information files.

**Funding:** The author(s) received no specific funding for this work.

**Competing interests:** The authors have declared that no competing interests exist.

## Introduction

Chronic kidney disease (CKD) constitutes a significant public health issue across the globe [1]. According to the KDIGO 2024 guidelines, CKD is defined as abnormalities in kidney structure or function that persist for more than 3 months and have implications for health. CKD is diagnosed if either of the following is present for more than 3 months: a glomerular filtration rate (GFR) less than 60 ml/min/1.73 $m^2$ or markers of kidney damage, which include albuminuria (albumin excretion rate ≥30 mg/g or ≥3 mg/mmol), abnormalities in urine sediment, electrolyte imbalances or other issues caused by tubular disorders, structural abnormalities detected by imaging, histological abnormalities from biopsy, or a history of kidney transplantation [2]. It increases the risk of mortality and morbidity in patients having renal failure by increasing a cardiovascular burden [3].

The prevalence of UN ranges between 60 and 100% in adult patients with CKD G5 when estimated GFR drops below 15 mL/min per 1.73 $m^2$ [2]. However, the prevalence is unknown in children and teenagers [4]. Although, clinical symptoms and objective data raise the possibility of UN, nerve conduction investigations using Electroneuromyography (EMG) are typically necessary for a definitive diagnosis, as well as for prognosis and the choice of therapy [5].

The diagnosis of UN in children is often made retrospectively when symptoms improve after dialysis, as there are no defining signs or specific laboratory and imaging findings [6]. Recent studies have shown that subclinical uremic polyneuropathy (PN) is seen in up to one-third of children with CKD G3 and above, indicating that the entity is probably unrecognized in children. The predominant type of neuropathy seen in this category of patients is controversial and there is a lack of data concerning the association between dialysis and UN in the pediatric population [7,8].

The aim of the study was to determine the prevalence of UN in children with ESRD in a Tunisian cohort. The second objective was to assess the effect of dialysis on UN and associated risk factors over a six-month period.

## Methodology

### Study design

A prospective cohort study was conducted between 20 July 2021 and 10 September 2023, involving children with ESRD followed in Department of Pediatric Nephrology at University Hospital of Sahloul (Tunisia).

### Study population

We included children aged 5–18 years who were diagnosed with CKD G5 defined by a GFR below 15 mL/min/1.73 $m^2$ [2] regardless of whether they were undergoing dialysis. Patients were included regardless of the presence of symptoms or signs of peripheral neuropathy (PN). However, we did not include patients with conditions known to impair central or peripheral nerve function (e.g., diabetes, hypothyroidism), CKD secondary to metabolic disorders (e.g., primary hyperoxaluria, cystinosis), and those whose parents did not provide consent.

## Study groups

Patients were divided into two distinct groups:

- Dialysis group: (exposed patients) Children who were receiving renal replacement therapy (RRT), including hemodialysis or peritoneal dialysis.

- Non-dialysis patients (non-exposed) Children who were not undergoing dialysis and were receiving conservative treatment for CKD G5.

## Sample size

To estimate the required sample size a priori, we used the Epi info version 6 software assuming a prevalence of 22% for PN [8], with 95% confidence interval and 5% precision level, giving an adjusted estimated sample size of 132 patients.

Given that the total eligible population of children with ESRD in Tunisia was estimated at N = 170 [9]. We applied the finite population correction. However, because of stringent criteria not including children with neuropathy from other causes—and limited recruitment feasibility, only 31 children were enrolled over two years.

## Sampling

At **T1** (baseline), all participants underwent EMG assessments.

Over the **6-month follow-up**, 7 patients were lost to follow-up (6 from the dialysis group and 1 from the pre-dialysis group). Additionally, 10 patients from the pre-dialysis group transitioned to dialysis. At **T2** (6-month follow-up), 17 patients were in the dialysis group, including those who transitioned from the pre-dialysis group, while 7 patients remained in the non-dialysis group (Fig 1).

## Ethical consideration

The study adhered to the Helsinki Declaration's Ethical Principles for Medical Research Involving Human Subjects. The study protocol received approval from the Regional Ethics Committee (Reference: CEFMS 97/2021). Informed oral consent was obtained from all parents or legal guardians, in keeping with ethics committee guidelines, using plain language suitable for low-literacy populations. Patients were free to participate in, decline, or withdraw from the EMG study at any time. The consent process was witnessed and documented to ensure full disclosure and voluntariness Parents/guardians were informed that participation was voluntary and that they could withdraw their child from the study at any time without consequence.

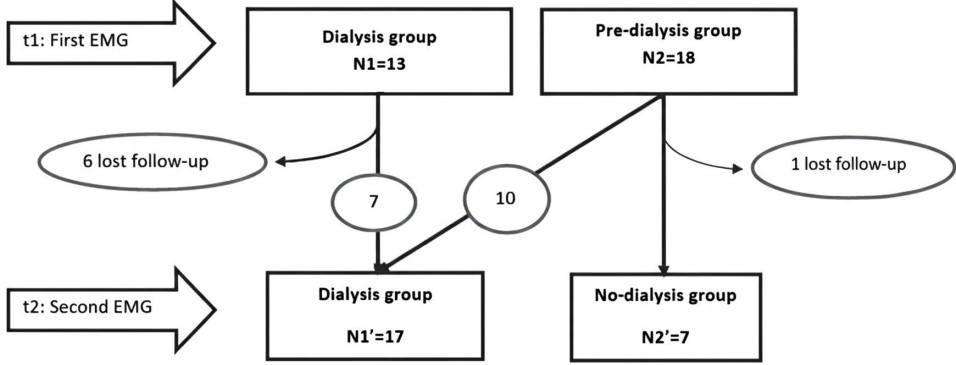

**Fig 1. Participants' follow-up between T1 and T2.**

## Procedure

We collected data for all patients using a survey, medical files, clinical examination, biological analysis and electromyographic studies. All variables were defined a priori.

PN was assessed through neurologic examination. Clinical neuropathy was assessed using a multimodal method that included a patellar reflex test using a reflex hammer. Muscle strength in children was assessed using the MRC (Medical Research Council) scale ranging from 0 to 5 [10], both deep and superficial sensitivity were carefully investigated. Neuropathic pain was evaluated using the DN4 score [11].

Superficial sensitivity examination followed a proximal-to-distal pattern along dermatomes, utilizing light touch with a finger. Profound sensitivity assessment involved testing position sense, requiring patients to identify the position of the big toe with closed eyes [12].

Clinical peripheral neuropathy was defined if at least one sensory and/or motor symptom or sign was found to be relevant during the clinical examination [13]. The GFR was determined for all patients. $\text{GFR } (\text{mL/min}/1.73 \text{ m}^2) = 36.5 \times height\,(cm)\,\frac{Height(cm)}{\text{creatinine }\mu\text{mol/L}}$ [14]. Normal Biological values for ESRD children were defined according to the guidelines outlined by KDIGO [15,16]. Weight-for-age, height-for-age, and BMI-for-age z-scores were calculated using WHO growth standards. Z-scores between −2 and +2 were considered normal [17]. To assess dialysis adequacy in hemodialysed patients, we relied on two parameters: Kt/V (urea clearance (K), dialysis duration (t), and urea distribution volume (V)) and Percent Reduction of Urea (PRU) [18].

## The electromyographic study

The EMG study was performed by an experienced neurophysiologist for all participants, using a Neurosoft® machine. The nerve conduction study (NCS) was performed according to the guidelines of American Association of Neuromuscular & Electrodiagnostic Medicine for children [19]. To ensure patient comfort, and since UN slowly progresses proximally, starting from the lower limbs and may spread to upper extremities [20]. EMG study focused on two motor nerves in the lower limb (tibial and peroneal nerves) with limited sensory studies to the sural and peroneus superficialis nerves [5]. Nerve Conduction Velocity (NCV), amplitude, distal latency, and F-wave were recorded, comparing all measures with age-adjusted reference values [21].

To avoid any risk of nociceptive fibers activation, the stimulation intensity used was always under 25 mA for motor nerve and 15 mA for sensory nerve [22].

The amplitude of Compound Muscle Action Potential (CMAP) was measured from baseline to peak. The amplitude of sensory nerve action potential (SNAP) was measured from peak to peak [21]. The conduction distal velocity and distal latency of the tested motor and sensory of nerves was recorded. In addition, the proximal motor velocity was measured with the F wave. All nerve conduction studies measures were compared with age adjusted reference values [23].

PN was diagnosed according to the electrophysiological Criteria for Axonal and Demyelinating Neuropathies [24]. The severity of PN was categorized based on the number of NCS abnormalities as follows: 0–1 abnormalities indicate mild severity, 2 abnormalities indicate moderate severity, and 3–4 abnormalities indicate severe severity [25]. Improvement in EMG parameters was defined by a reduction in neuropathic severity or normalization of previously abnormal findings.

## Statistical study

Data were analyzed using IBM SPSS Statistics version 22.0 (SPSS Inc., Chicago, IL, USA). Normal distribution and variance equality were assessed using the Kolmogorov-Smirnov test. Descriptive analysis included mean, standard deviation, minimum, and maximum values. For asymmetric distributions, the median and the 25th and 75th percentiles were used. Confidence intervals of 95% were calculated. Absolute and relative frequencies were used for qualitative variables. For quantitative data, comparisons between two means were made using the Student t-test. The Mann-Whitney test was used

for non-normally distributed variables. The Chi-squared test, Pearson test, and Fisher's exact test were used for qualitative data comparison. P<0.05 was considered significant.

## Results

### Clinical and Biological characteristics

The study population had a mean age of 11±3.5 years, with CKD diagnosed at a mean age of 7.5±4.2 years (range: 5–18 years). The male-to-female ratio was 1.06. At enrollment, the mean weight-for-age was −0.84 (SD±1.2; range: −2.7 to 2.4), and the mean height-for-age was −1.69 (SD±1.7; range: −5.6 to 1.6). BMI-for-age had a mean of −0.4 S (SD±1.2; range: −3.22 to 1.83), with 10% of the population classified as underweight (z-score BMI-for-age<−2). All patients had CKD G5, with a mean duration of 54.1±45.4 months at the time of EMG. The leading cause of CKD was congenital abnormalities of the kidney and urinary tract (CAKUT) (58.1%), followed by glomerular nephritis (12.9%) and hereditary nephritis (9.7%).

Only 13 patients were under RRT. All patients treated by hemodialysis (HD) were on conventional HD (Three sessions per week) with standardized parameters dialysis, with an average dialysis session recovery time of 3–4 hours [26]. Neurological signs of neuropathy were found in 32% of patients. The biological analysis between the 2 groups didn't find a statistical difference in terms of biological parameters (Table 1).

Table 1. Comparison between pre-dialysis and dialysis groups regarding clinical features and biological findings.

| Clinical Features | Pre-dialysis (N=18) | Dialysis (N=13) | p-value |
|---|---|---|---|
| Age (years)[a] | 10.89±3.03 | 11.77±4.19 | 0.5 |
| Height for age[a] | (−1.61) ± 1.90 | (−1.80) ± 1.59 | 0.7 |
| Weight for age[a] | (−0.63) ± 1.37 | (−1.15) ± 0.98 | 0.2 |
| BMI for age (kg/m$^2$)[a] | (−0.09) ± 1.2 | (−0.9) ± 1.2 | 0.1 |
| Clinical neuropathy[c] | 6 (33%) | 4 (30%) | 1.0 |
| Time of evolution of kidney disease[a] | 52.8±46.28 | 32.2±32.2 | 0.1 |
| Age of diagnosis of kidney disease[a] | 9.23±4.58 | 10.89±2.55 | 0.1 |
| **Biological Findings** | | | |
| Urea (mmol/L)[a] | 20.6±11.7 | 27.8±14.9 | 0.3 |
| Serum creatinine (µmol/L)[a] | 539.8±306.8 | 491.6±313.7 | 0.3 |
| GFR (mL/min/1.73m$^2$)[a] | 8.98±4.26 | 11.78±6.45 | 0.1 |
| Potassium (mmol/L)[a] | 4.3±0.9 | 4.1±0.9 | 0.7 |
| Calcium (mmol/L)[a] | 2.37±0.27 | 2.29±0.31 | 0.5 |
| Phosphorus (mmol/L)[a] | 1.61±0.47 | 2.41±1.53 | 0.1 |
| 25-hydroxy-vitamin D (ng/mL)[b] | 15.15 [10.6-27.05] | 26 [13.1-33.6] | 0.1 |
| Alkaline phosphatase (U/L)[b] | 287 [181.5-436.5] | 255 [190-468] | 0.9 |
| Albumin (g/L)[a] | 33.18±5.66 | 34.90±7.11 | 0.6 |
| Hemoglobin (g/dL)[a] | 9.1±2.4 | 8.8±1.8 | 0.6 |
| Parathyroid hormone (PTH) (pg/mL)[b] | 196.9 [140.2-982.5] | 224 [106-360.7] | 0.6 |
| Ferritin level (ng/mL)[b] | 232 [95.4-604] | 259 [161.5-584] | 0.9 |

[a]Quantitative data with normal distribution were means±standard deviation.

[b]Quantitative data with no normal distribution were median (IQR).

[c]Qualitative data were number (%). Chi2 test was used to compare qualitative data.

Student t test was used to compare quantitative data with normal distribution. Mann Whitney U test was used to compare quantitative data with no normal distribution.

p<0.05 was statistically significant.

## First EMG analysis

The first EMG was pathological in 45% of cases across all patients, with no statistical difference between the two groups. Silent neuropathy, defined as pathological EMG results without clinical neuropathy, had a prevalence of 12.9%.

In all cases, PN exhibited an axonal mechanism, with no instances of demyelinating PN. NCS has showed that the tibial nerve is the most affected.

Remarkably, F-wave in the tibial nerve was absent or prolonged in 32% of patients with PN. Although, none of the observed variations reached statistical significance between the two groups (Table 2).

## Second EMG

The study found that in the dialysis group, 88.8% showed improvement in pathological EMG results, unlike the no-dialysis group, which showed no improvements (Table 3).

We compared the EMG outcomes of 17 patients on dialysis at T2, including 10 who were not on dialysis at T1. The second EMG assessment at T2 showed improved outcomes, with more normal cases and complete resolution of severe abnormalities (Fig 2).

**Table 2. Comparison of electrophysiological abnormalities between pre-dialysis and dialysis Groups among children (n = 31) having an End Stage Kidney Disease.**

|  | Total[a] (n = 31) N (%) | Pre-dialysis group[a] (n = 18) N (%) | Dialysis group[a] (n = 13) N (%) | p-value |
|---|---|---|---|---|
| **Presence of PN** | 14 (45) | 9 (50) | 5 (38.5) | 0.5 |
| **Neuropathy Type** | | | | |
| Motor neuropathy | 6 (19.3) | 4 (42.9) | 2 (15.4) | 0.8 |
| Sensory neuropathy | 2 (6.4) | 1 (5.5) | 1 (8.3) | |
| Sensitivomotor neuropathy | 5 (16) | 3 (16.7) | 2 (15.4) | |
| **Peroneal Nerve** | | | | |
| Prolonged mdl | 2 (6.4) | 2 (11.1) | 0 (0) | 0.1 |
| Reduced CMAP | 5 (16) | 5 (27.8) | 0 (0) | 0.1 |
| Reduced MCV | 6 (19.3) | 5 (27.8) | 1 (8.3) | 0.3 |
| **Tibial Nerve** | | | | |
| Prolonged mdl | 3 (9.6) | 3 (16.7) | 0 (0) | 0.6 |
| Reduced CMAP | 7 (22.5) | 4 (42.9) | 3 (23.1) | 1.0 |
| Reduced MCV | | – | – | – |
| **Sural nerve** | | | | |
| Reduced CMAP | 7 (22.5) | 4 (22.2) | 3 (23.1) | 0.9 |
| Reduced MCV | 7/31 (22.5) | 4 (22.2) | 3 (23.1) | 0.9 |
| **Superficial peroneal nerve** | | | | |
| Reduced CMAP | 4 (13) | 2 (11.1) | 2 (15.4) | 1.0 |
| Reduced MCV | 41 (13) | 2 (11.1) | 2 (15.4) | 1.0 |
| **F-wave** | | | | |
| Prolonged/Absent F-wave (Tibial) | 10 (32%) | 7 (38.9%) | 3 (23.1%) | 0.6 |

[a]Qualitative data were number (%). Chi2 test was used to compare qualitative data.

p<0.05 was statistically significant.

**Table 3. Evolution of the pathological EMG in dialysis and no-dialysis group.**

|  | No improvement | Improvement | % Improvement | P |
|---|---|---|---|---|
| **No-dialysis group** | 3 | 0 | 0% | **0.01** |
| **Dialysis group** | 1 | 8 | 88.89% | |

Chi2 test was used to compare qualitative data.

p<0.05 was statistically significant.

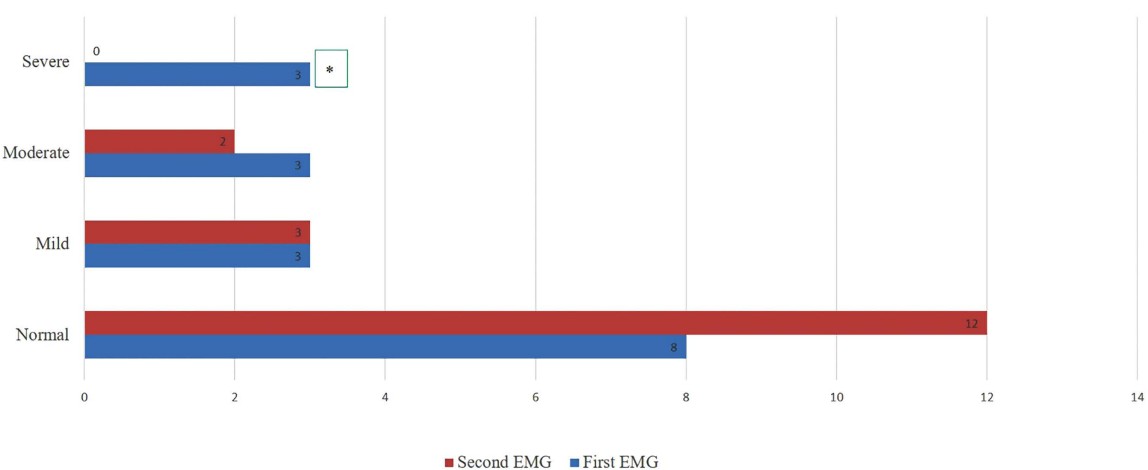

**Fig 2. Comparison of Severity Grade and Normal EMG Between First and Second EMG Assessments in dialysis group at T2.** Chi2 test was used to compare qualitative data. * p<0.05 was statistically significant.

Significant improvements were observed in CMAP amplitude for the peroneal and tibial nerves, SNAP amplitude for the sural nerve, and sensory NCV for the superficial peroneal and sural nerves in the dialysis group, while no significant changes were noted in the no-dialysis group (Table 4).

## Factors associated with the outcome of uremic neuropathy

In our patient cohort, the presence of clinical neuropathy, an underlying glomerular nephropathy and weight-for-age were found to be correlated with the presence of PN.

The subset exhibiting pathological EMG results appeared to demonstrate higher serum creatinine, Parathyroid hormone (PTH) and ferritin levels along with lower GFR. These differences did not reach statistical significance (p>0.05). In the group of dialysis patients, we found no association between the KT/V and the PRU and the UN (Table 5).

## Factors associated with EMG improvement

The improvement of UN on EMG parameters was only associated with the dialysis (Table 6). The results suggest that while there is a trend indicating that patients with moderate-severe PN are more likely to experience improvement in EMG compared to those with mild PN, this difference is not statistically significant (p=0.22). Dialysis was considered as a protective factor. As for biological parameters. The change in urea levels (T2-T1) was the only biological variable linked to EMG improvement, with a non-significant Levene's test (p=0.093) confirming equal variances and a significant t-test result (p=0.049) (Table 6).

**Table 4. Evolution of EMG parameters between first and second EMG among children having an End Stage Kidney Disease with an Uremic neuropathy (n = 24).**

| | Groups | Difference in Mean | p-value |
|---|---|---|---|
| **Peroneal nerve** | | | |
| Onset2 –Onset1 (ms) | Dialysis group (n = 17) | 0.37 | 0.1 |
| | No-dialysis (n = 7) | 0.25 | 0.2 |
| Amplitude2- Amplitude1(mV) | Dialysis group | 1.39 | **<0.05** |
| | No-dialysis | −0.28 | 0.1 |
| NCV2- NCV1(m/s) | Dialysis group | – | 0.2 |
| | No-dialysis | – | 0.1 |
| Tibial nerve | | | |
| Onset2 –Onset1 (ms) | Dialysis group | 0.62 | 0.2 |
| | No-dialysis | −0.05 | 0.8 |
| Amplitude2- Amplitude1 (mV) | Dialysis group | 1.87 | **<0.05** |
| | No-dialysis | 1.87 | 0.7 |
| **Superficial peroneal** | | | |
| Amplitude2- Amplitude1 (µV) | Dialysis group | 8.76 | 0.1 |
| | No-dialysis | −9.78 | 0.3 |
| NCV2- NCV1 (m/s) | Dialysis group | – | **<0.05** |
| | No-dialysis | 6.23 | 0.40 |
| **Sural nerve** | | | |
| Amplitude2- Amplitude1 (µV) | Dialysis group | 6.15 | **<0.05** |
| | No-dialysis | 0.35 | 0.9 |
| NCV2- NCV1(m/s) | Dialysis group | – | 0.2 |
| | No-dialysis | −2.53 | 0.6 |

Paired t-test was used.

p < 0.05 was statistically significant.

## Discussion

The mean aim of our study was to determine the prevalence of UN in children with ESRD and to examine the association between dialysis treatment and the outcome of UN. Thirty-one children diagnosed with ESRD, were enrolled in our six-month study and underwent regular EMG assessments to identify the presence of uremic neuropathy. The prevalence of UN was present in 45% of our study population with a high prevalence of silent neuropathy. The neuropathy was predominantly characterized by a high incidence of axonal involvement, particularly in its pure motor and sensorimotor forms, with the tibial nerve most frequently affected. No significant differences were observed in UN characteristics between children who received dialysis and those who did not. The presence of UN in EMG was associated to clinical neuropathy, lower weight-for-age scores, and underlying glomerular nephritis. Over the follow-up period, UN showed noteworthy improvement, and this recovery was correlated with dialysis treatment.

Knowledge about the prevalence of uremic neuropathy (UN) in children remains limited, as the few existing studies reported widely varying rates ranging from 22 to over 50% [7]. This variability could be attributed to differences in diagnostic criteria and study populations [8]. Additionally, the silent neuropathy was described in 13% among all patients. Some studies reported 29% of children with CKD G3 and above presented sub-clinical neuropathy detected by EMG. The clinical symptoms and the electrophysiologic indicators often diverge. The main mechanism is that UN favors large-diameter axons while preserving both tiny and unmyelinated neurons, involved in thermoalgesic sensitivity and neuropathic pain [27].

**Table 5. Association between polyneuropathy presence and clinical/biological features during initial EMG in a 31-patient cohort.**

| | Normal n = 17 | Polyneuropathy n = 14 | P |
|---|---|---|---|
| Age (In year)[a] | 10.19 ± 3.3 | 12.4 ± 3.4 | 0.1 |
| Male Gender [b] | 9 (52.9) | 7(50) | 0.59 |
| Consanguinity[b] | 11(64.7) | 10(71.4) | 0.6 |
| Dialysis[b] | 7(41.2) | 6 (42.9) | 0.8 |
| Weight for age[a] | (−1.2DS) ±1.7 | (−0.3DS)±1.2 | **0.03** |
| BMI for age[a] | (−0.6DS)± 1 | (−0.17DS) ± 1.4 | 0.2 |
| nPCR(g/kg/day) [a] | 4.4 ± 1.01 | 3.4 ± 1.1 | 0.2 |
| Height for age[a] | (−2.1DS)±1.8 | (−1.1DS)±1.4 | 0.1 |
| Duration of the CKD[c] | 24 [12-72] | 42 [9.7-90] | 0.4 |
| Etiologies of CKD[b] | | | |
| Glomerular nephritis | 0 (0) | 4 (100) | **0.03** |
| CAKUT | 12 (70) | 6 (42) | 0.1 |
| Others | 4 (23) | 5 (35) | 0.9 |
| Clinical neuropathy[b] | 0 (0) | 10 (71.4) | **<0.001** |
| Serum creatinine (µmol/L)[a] | 506.8 ± 245.4 | 563.8 ± 355.8 | 0.6 |
| Urea (mmol/l)[a] | 23.6 ± 8.3 | 27.5 ± 17.7 | 0.4 |
| GFR (ml/min/1.73m2)[a] | 11.1 ± 5.7 | 9.9 ± 5.7 | 0.5 |
| Kalemia(mmol/L)[a] | 4.2 ± 1.03 | 4.07 ± 0.7 | 0.2 |
| Calcium (mmol/L)[a] | 2.39 ± 0.2 | 2.26 ± 0.33 | 0.2 |
| Phosphorus (mmol/L)[a] | 1.7 ± 0.3 | 2.4 ± 1.8 | 0.1 |
| Alkaline phosphatase (U/L)[b] | 255 [200-461] | 287 [153-465] | 0.8 |
| 25-hydroxy-vitamin D (ng/mL)[b] | 23.1 [10.6-35.2] | 26 [11.4-31] | 0.9 |
| Albumin (g/L)[a] | 37.12 ± 4.2 | 30.8 ± 7.2 | 0.1 |
| Hemoglobin (g/dL)[a] | 9.5 ± 1.9 | 8.37 ± .2.16 | 0.1 |
| PTH (pg/mL)[b] | 193.5 [143.5-687] | 248 [69.4-373] | 0.6 |
| Ferritin level[a] (ng/mL)[b] | 232.7 [80-553] | 391.9 [123.2-719.2] | 0.4 |
| KT/V (dialysis group)[b] | – | 1.6 [1.4-2.2] | 0.7 |
| PRU [a](dialysis group) | 0.76 ± 0.03 | 0.72 ± 0.08 | 0.48 |

[a]Quantitive data with normal distribution were means ± standard deviation.

[b]Quantitative data with no normal distribution were median (IQR).

Student t test was used to compare quantitative data with normal distribution.

Mann Whitney U test was used to compare quantitative data with no normal distribution.

The most electrophysiological pattern observed in our study was sensorimotor and motor neuropathy. Our results are similar to those of Yoghantan et al showing that primary motor axonal neuropathy is the principal pattern observed [7]. Some others studies reported an axonal motor and sensory neuropathy, whereas demyelinating motor neuropathy was less common [8].

The electrical differences in UN between adults and pediatric patients suggest distinct pathophysiological mechanisms. In adults, UN is often associated with diabetes, while in children, it is frequently linked to malformative origins of ESRD [1].

Our study showed that the tibial nerve is the most affected. These results are in agreement with previous studies like Yoganathan et al.,showing the most commonly affected nerves, according were the tibial and common peroneal nerves [7].

**Table 6. Factors associated with EMG improvement after 6 months of follow-up of the uremic neuropathy in patients having a pathological first EMG.**

| | Improvement (N = 8) | No improvement (N = 4) | RR [CI] | p-value |
|---|---|---|---|---|
| **Gender[c]** | | | | |
| Male | 3(37) | 3(75) | | 0.5 |
| Female | 5(63) | 1(25) | | |
| Age (Mean ±SD)[a] | 12.3±3.6 | 12.2±4.5 | | 0.9 |
| Consanguinity[c] | 6(85) | 2(50) | | 0.1 |
| Weight-for-age[a] | −0.22±1.6 | −0.25±0.6 | | 0.9 |
| Height-for-age[a] | −1.2±1.6 | −0.9±1.5 | | 0.8 |
| BMI-for-age[a] | −0.1±1.7 | 0.3±0.6 | | 0.6 |
| Duration of the CKD[b] | 48.5 [9.7-102] | 47.5 [17.7-96] | | 0.5 |
| **Etiologies of CKD[c]** | | | | |
| Glomerular nephritis | 3(37.5) | 0(0) | | 0.4 |
| CAKUT | 4(50) | 3(75) | | 0.5 |
| Others | 1(12.5) | 1(25) | | 0.8 |
| Dialysis[c] | 8(100) | 1(25) | **9 [1.4-57.1]** | **<0.05** |
| Clinical neuropathy[c] | 6(75) | 3(75) | | 1.0 |
| KT/Va (dialysis group)[b] | 1.7 [1.4-2.2] | 0.95 | | 0.6 |
| PRU (dialysis group)[b] | 0.7±0.07 | 0.59 | | 0.1 |
| **PN severity[c]** | | | | |
| Mild PN | 2(25) | 3(75) | | 0.2 |
| Moderate-Severe | 6(75) | 1(25) | | |
| **Type of PN[c]** | | | | |
| Pure motor | 2(25) | 2(50) | | 0.5 |
| Sensitivo-motor | 4(50) | 1(25) | | 0.5 |
| Other | 2(25) | 1(25) | | 1.0 |

[a]Quantitaive data with normal distribution were means±standard deviation.

[b]Quantitative data with no normal distribution were median (IQR).

[c]Qualitative data were number (%).

Chi2 test was used to compare qualitative data. Student t test was used to compare quantitative data with normal distribution. Mann whitney U test was used to compare quantitative data with no normal distribution. Levene's Test (p): Tests for equality of variances between groups. A p<0.05 indicates unequal variances. t-Test (p): Compares the means of differences between groups. A p<0.05 indicates a significant difference.

Ackil et al had observed absent F-waves in 59% of children. In the present study, F-wave latencies were studied in the tibial nerve only, this possibly can explain the fewer F-wave abnormalities [28].

In our study, we found significant improvements in CMAP amplitude for the peroneal and tibial nerves, as well as sensory NCV for the superficial peroneal nerve in the dialysis group.

However, patients who never underwent dialysis did not show any significant electrophysiological improvement. A previous study conducted showed that following dialysis, electrodiagnostic tests showed an increase in sensory and motor evoked wave amplitudes, indicating activation of previously inactive axons. While nerve conduction velocity remained relatively unchanged, the rise in wave amplitude suggests a reversal of axonal inactivation, possibly due to the removal of accumulated toxins during HD [29].

In our patient group, clinical neuropathy, glomerular nephropathy, and weight-for-age were associated with PN, while age, height, and gender showed no association with UN.

Although our study did not find age to be linked with UN, previous research has shown an association with older age [7].

A pathological neurological clinical exam was significantly associated with UN. Sameh A. et al. and Yoganthan et al. identified clinical neuropathy, such as hyporeflexia and numbness, as factors associated with UN in children [7,8].

It has been confirmed in previous studies that malnutrition is associated to small fiber neuropathy among patients maintained on HD. Malnutrition in dialysis patients has been linked to elevated levels of tumor necrosis factor-α (TNF-α), therefore, malnutrition risk could potentially contribute to the development of small fiber neuropathy in this patient population by exacerbating inflammatory and oxidative stress pathways [30].

Yoganthan et al. identified factors significantly associated with PN, including underlying glomerular disease suggesting a possible common immunopathogenesis in the association of membranous glomerulonephritis and inflammatory demyelinating peripheral neuropathies [7].

Our research findings indicated no notable associations between biological values and UN. However, a prior study suggested an increased prevalence of neuropathy in patients with lower GFR and higher creatinine and albumin levels [8]. Yoganthan et al. identified low albumin, low copper levels, and high serum ferritin to be associated with UN [7]. A direct correlation was proved between serum potassium levels and neurophysiological parameters, a study revealed that even upon normalization of other dialyzable compounds, axonal depolarization persists as long as potassium levels remain elevated. This suggests that pre-dialysis axonal depolarization is primarily influenced by serum potassium, casting doubt on the significant involvement of middle molecule toxins in this dysfunction [31].

The effect of PTH on nerve conduction studies in patients with CKD varied among studies. Despite previous findings by Massry and Goldstein and by Avram et al., which did not establish a relationship between PTH levels and PN [32], several studies suggested an association with hyperparathyroidism [33].

In our study, dialysis therapy emerged as the only influential factor in determining EMG improvement. Early investigations into the effects of HD on UN indicated that UN can be minimized by early dialysis or reversed by renal transplantation [34].

Long-term adequate dialysis, lasting more than a year, is associated with a significant increase in the motor-nerve conduction velocities and improvement of clinical neuropathy [35]. In contrast, other studies have shown that improvement in neuropathy with dialysis is rare. Comparative studies between HD and Peritoneal dialysis have not shown a significant difference in neuropathy progression.

Indices of dialysis efficiency (Kt/V, PRU) were calculated for HD patients, our results showed no association between KT/V, PRU and UN. Janda et al. reported no significant correlation between indices of dialysis efficiency (Kt/V, PRU) and nerve conduction velocity in the tested nerves [36].

Recent research suggests that there is little improvement in neuropathy, particularly in severe cases and those with a motor-predominant pattern, with dialysis [34].

## Limits of the study

To the best of our knowledge, this study is the first to examine UN in children through a prospective design. To ensure the robustness of our study, we aimed to calculate the sample size and adjusted it. However, due to strict exclusion criteria (excluding children with neuropathy from other causes) and limited recruitment feasibility, the final enrolled cohort consisted of 31 children over 2 years. This number reflects the practical constraints of a rare disease setting, the high burden of comorbidities excluding eligibility, and ethical considerations in paediatric research. Importantly, our study retains value by reporting prevalence estimates with appropriate confidence intervals and serving as a first prospective study of this condition in this population.

Additionally, since this study design is susceptible to loss to follow-up, we experienced seven such cases: two children died, four became unreachable by phone, and one received a renal transplant. Besides, many children were not included in the study to avoid any systemic condition that could cause PN.

Otherwise, possible biological factors predicting neuropathy in patients with CKD that were not assessed such as β-2 microglobulin, myoinositol, guanidinosuccinic acid, methylguanidine, polyamines, phenol derivatives, biotin, pyridoxine,

cobalamin, zinc, glycosylated hemoglobin, vitamin B1, and B6 levels due to the unavailability of specific reagents in the laboratory [27]. The EMG is the gold standard exam for diagnosing PN. Simplified protocol was employed in our study to limit pain and discomfort in children [24]. As the used technique was relatively painful and complete, neurophysiologic studies were missing in some cases. The EMG study in children differ from adults by the importance of respecting the comfort of the patients and minimizing the number of nerves studied [21]. There is no clear consensus regarding the classification of electrophysiological severity in PN among pediatric population. To address this, we drew inspiration from others studies and from our practice in our laboratory [25]. Needle study was not performed which represent a limitation in our study [27].

## Conclusion

This study is one of the few investigating UN in children through a prospective design. Our research indicates that UN is predominant in children, with a high incidence of the silent form. Electromyographic monitoring showed that dialysis was associated with an improvement in UN. Our findings could be used further to confirm the role of EMG as a screening tool for children with ESRD and a way to assess treatment in these patients.

## Acknowledgments

We express our gratitude to the dialysis unit staff for their support and to the patients and their parents for their cooperation.

## Author contributions

**Conceptualization:** Nouha Gammoudi.

**Data curation:** Arwa Yahyaoui, Hela Ghali.

**Formal analysis:** Arwa Yahyaoui, Nouha Gammoudi, Hela Ghali.

**Investigation:** Nouha Gammoudi, Selsabil Nouir.

**Methodology:** Arwa Yahyaoui, Nouha Gammoudi, Selsabil Nouir, Sameh Mabrouk.

**Resources:** Selsabil Nouir, Sameh Mabrouk.

**Supervision:** Saoussen Abroug, Ghazi Sakly.

**Writing – original draft:** Arwa Yahyaoui, Nouha Gammoudi, Selsabil Nouir.

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
