## [Decision Letter · Decision Letter 0]

16 Jun 2025

Dear Dr. Yahyaoui,

Thank you for submitting your manuscript to PLOS ONE. After careful consideration, we feel that it has merit but does not fully meet PLOS ONE’s publication criteria as it currently stands. Therefore, we invite you to submit a revised version of the manuscript that addresses the points raised during the review process.

**ACADEMIC EDITOR: ** Please respond reviewer's concerns.

We look forward to receiving your revised manuscript.

Kind regards,

Ken Iseri

Academic Editor

PLOS ONE

Journal Requirements:

Additional Editor Comments:

Please respond reviewer's concerns.

Reviewers' comments:

Reviewer's Responses to Questions

**Comments to the Author**

1. Is the manuscript technically sound, and do the data support the conclusions?

Reviewer #1: Partly

Reviewer #2: No

2. Has the statistical analysis been performed appropriately and rigorously?

Reviewer #1: I Don't Know

Reviewer #2: No

3. Have the authors made all data underlying the findings in their manuscript fully available?

Reviewer #1: Yes

Reviewer #2: No

4. Is the manuscript presented in an intelligible fashion and written in standard English?

Reviewer #1: Yes

Reviewer #2: No

Reviewer #1: Hello

This is a good article

The topic of the article is also interesting and new.

The topic of neuropathy in children and kidney diseases is very important and can have a great impact on the prevention and treatment of the disease.

Good luck.

Reviewer #2: Comments and reviewing points for the Author/Editor

1- The text of the introduction and the title are weakly consistent.

2- The type of study was not chosen well.

4- The study method has many shortcomings and is not acceptable.

5- The sample size was determined incorrectly and is too small.

6- Due to the small sample size, both most of the statistical calculations performed and the results presented are incorrect.

7- The discussion section is very poorly written.

**Do you want your identity to be public for this peer review?** For information about this choice, including consent withdrawal, please see our Privacy Policy

Reviewer #1: No

Reviewer #2: No

---

## [Author Response · Author response to Decision Letter 1]

25 Aug 2025

Editor Comments:

Formatting and Style: We have revised the manuscript to comply with PLOS ONE’s style requirements, including the title page, figure formatting, and file naming conventions.

Data Availability Statement: All relevant data are available upon request from the corresponding author.

Reviewer #1:

General Comment: Reviewer #1 provided positive feedback on the novelty and relevance of the topic. We greatly appreciate their encouraging comments and have made no further changes in response to their feedback.

Reviewer #2:

Introduction and Title: We revised the introduction to better align with the clinical context and study objectives. The title has been updated to better reflect the scope of the study:

“Prevalence of uremic neuropathy and the effect of dialysis in children with end-stage renal disease: a cohort study.”

Study Design: We clarified our choice of a prospective cohort study and explained how it was designed to assess the prevalence and progression of uremic neuropathy in children with ESRD.

Methodology: We added a sampling section to better define inclusion criteria and clarified the nerve conduction study protocol and diagnostic methods.

Sample Size: We included a more detailed explanation of the sample size calculation, noting the practical constraints in recruiting pediatric ESRD patients and the rare nature of the disease.

Statistical Analysis: We described the statistical methods used for small sample sizes, clarified the tests applied, and acknowledged the limitations due to the sample size.

Discussion: The Discussion section has been rewritten to improve clarity and structure, including a summary of key findings, comparison with literature, and discussion of clinical implications and limitations.

---

## [Decision Letter · Decision Letter 1]

14 Oct 2025

Dear Dr. Yahyaoui,

Thank you for submitting your manuscript to PLOS ONE. After careful consideration, we feel that it has merit but does not fully meet PLOS ONE’s publication criteria as it currently stands. Therefore, we invite you to submit a revised version of the manuscript that addresses the points raised during the review process.

**ACADEMIC EDITOR: ** Apologies for the time taken to review.　Some minor adjustments are required, please find  the reviewer's comment.

We look forward to receiving your revised manuscript.

Kind regards,

Ken Iseri

Academic Editor

PLOS ONE

Journal Requirements:

Reviewers' comments:

Reviewer's Responses to Questions

**Comments to the Author**

Reviewer #3: All comments have been addressed

2. Is the manuscript technically sound, and do the data support the conclusions?

Reviewer #3: Yes

3. Has the statistical analysis been performed appropriately and rigorously?

Reviewer #3: Yes

4. Have the authors made all data underlying the findings in their manuscript fully available?

Reviewer #3: Yes

5. Is the manuscript presented in an intelligible fashion and written in standard English?

Reviewer #3: Yes

Reviewer #3: The aim of the study was to determine the prevalence of UN in children with ESRD in a North

African cohort. The second objective was to assess the effect of dialysis on UN and associated

risk factors over a six-month period. Below please find my comments:

1. Authors indicate that the study aimed to assess UN in children ESRD in a North African population. The study was conducted in Tunisia, as far as I understand. Therefore, it would be better if the reference to North African population is removed everywhere and replaced by Tunisia as one of the North African countries.

2. Line 80 – please change to “Study Design”

3. In the Study Design section, please indicate where the study was conducted.

4. Line 85 – please define the Stage 5 chronic kidney disease, what are the criteria?

5. Line 107-110 – please define what T1 and T2 means.

6. In the ethics section, it is indicated that a consent was received from the patient or the legal guardian. The best practice is that the assent is received from the minor who is old enough. Since you had up to 18 year-olds, did you get assent (minor’s consent) or had only consent from the parent/legal guardian.

7. Line 180 – “-0.84 SD (±1.2; range: -2.7 to 2.4)” – I think it should be changed to -0.84 (SD ±1.2; range: -2.7 to 2.4). The same refers to other two SDs in the following sentences.

8. for the weight-for-age, height-for-age, BMI-for-age, indicate normal values.

9. Please define the abbreviation "PTH"

**Do you want your identity to be public for this peer review?** For information about this choice, including consent withdrawal, please see our Privacy Policy

Reviewer #3: No

---

## [Author Response · Author response to Decision Letter 2]

26 Oct 2025

Journal requirement:

The reference list was reviewed and updated to ensure accuracy and currency; no cited articles have been retracted.

Reviewer3: Comment 1: Authors indicate that the study aimed to assess UN in children ESRD in a North African population. The study was conducted in Tunisia, as far as I understand. Therefore, it would be better if the reference to North African population is removed everywhere and replaced by Tunisia as one of the North African countries.

Response: We thank the reviewer for this helpful clarification. We have replaced all mentions of “North African population” with “Tunisian population” throughout the manuscript.

Comment 2: Line 80 – please change to “Study Design.”

Response: Corrected as suggested.

Comment 3: In the Study Design section, please indicate where the study was conducted.

Response: We have added the location: “followed in Department of Pediatric Nephrology at University Hospital of Sahloul (Tunisia).”

Comment 4: Line 85 – please define Stage 5 chronic kidney disease, what are the criteria?

Response: We had already defined Stage 5 CKD in the Introduction (line 63-64) as “an eGFR < 15 mL/min/1.73 m² according to KDIGO 2024 criteria.” To improve clarity, we have now also repeated this definition briefly in the Study Design section (line 85) and updated the terminology to align with the KDIGO classification system (CKD G5, CKD G3..).

Comment 5: Lines 107–110 – please define what T1 and T2 mean.

Response: We added the clarification: “T1 refers to baseline assessment; T2 refers to the 6-month follow-up evaluation.”

Comment 6: In the ethics section, indicate whether assent from minors was obtained.

Response: Children were free to participate, decline, or withdraw from the EMG study at any time. We respected the child’s wishes, and any examination was immediately stopped if the child showed signs of discomfort or refusal. The EMG protocol was limited to a few nerves, and stimulation intensity was kept below 25 mA for motor nerves and 15 mA for sensory nerves to avoid activation of nociceptive fibers.

Comment 7: Line 180 – please adjust SD formatting.

Response: We corrected all SD expressions as suggested (now “-0.84 (SD ±1.2; range: -2.7 to 2.4)”).

Comment 8: For weight-for-age, height-for-age, BMI-for-age, indicate normal values.

Response: We have added reference ranges according to WHO growth standards (2007).

Comment 9: Please define the abbreviation “PTH.”

Response: Defined as “parathyroid hormone (PTH)” at first mention.

---

## [Decision Letter · Decision Letter 2]

12 Nov 2025

Prevalence of uremic neuropathy and the effect of dialysis in children with end-stage renal disease: a cohort study

PONE-D-25-17216R2

Dear Dr. Yahyaoui,

We’re pleased to inform you that your manuscript has been judged scientifically suitable for publication and will be formally accepted for publication once it meets all outstanding technical requirements.

Kind regards,

Ken Iseri

Academic Editor

PLOS ONE

Additional Editor Comments (optional):

Reviewers' comments:

Reviewer's Responses to Questions

**Comments to the Author**

Reviewer #3: All comments have been addressed

2. Is the manuscript technically sound, and do the data support the conclusions?

Reviewer #3: Yes

3. Has the statistical analysis been performed appropriately and rigorously?

Reviewer #3: Yes

4. Have the authors made all data underlying the findings in their manuscript fully available?

Reviewer #3: Yes

5. Is the manuscript presented in an intelligible fashion and written in standard English?

Reviewer #3: Yes

Reviewer #3: (No Response)

**Do you want your identity to be public for this peer review?** For information about this choice, including consent withdrawal, please see our Privacy Policy

Reviewer #3: No

---

## [Editor Report · Acceptance letter]

PONE-D-25-17216R2

PLOS ONE

Dear Dr. Yahyaoui,

I'm pleased to inform you that your manuscript has been deemed suitable for publication in PLOS ONE. Congratulations! Your manuscript is now being handed over to our production team.

Kind regards,

on behalf of

Dr. Ken Iseri

Academic Editor

PLOS ONE